# A fungal plant pathogen discovered in the Devonian Rhynie Chert

Christine Strullu-Derrien [1,2] ✉, Tomasz Goral[3,4], Alan R. T. Spencer [1,5], Paul Kenrick [1], M. Catherine Aime [6], Ester Gaya [7] & David L. Hawksworth [1,7,8,9]

*Fungi* are integral to well-functioning ecosystems, and their broader impact on Earth systems is widely acknowledged. Fossil evidence from the Rhynie Chert (Scotland, UK) shows that *Fungi* were already diverse in terrestrial ecosystems over 407-million-years-ago, yet evidence for the occurrence of *Dikarya (*the subkingdom of *Fungi* that includes the phyla *Ascomycota* and *Basidiomycota*) in this site is scant. Here we describe a particularly well-preserved asexual fungus from the Rhynie Chert which we examined using brightfield and confocal microscopy. We document *Potteromyces asteroxylicola* gen. et sp. nov. that we attribute to *Ascomycota incertae sedis (Dikarya)*. The fungus forms a stroma-like structure with conidiophores arising in tufts outside the cuticle on aerial axes and leaf-like appendages of the lycopsid plant *Asteroxylon mackiei*. It causes a reaction in the plant that gives rise to dome-shaped surface projections. This suite of features in the fungus together with the plant reaction tissues provides evidence of it being a plant pathogenic fungus. The fungus evidently belongs to an extinct lineage of ascomycetes that could serve as a minimum node age calibration point for the *Ascomycota* as a whole, or even the *Dikarya* crown group, along with some other *Ascomycota* previously documented in the Rhynie Chert.

The 407-million-year-old Rhynie Chert is a fossilized geothermal site and one of the most ancient known land plant deposits. Its exceptionally well-preserved fossils provide a unique view of an ancient terrestrial ecosystem that has been crucial to elucidating early land plant evolution (e.g., [1–3]). The deposit includes animals (mostly arthropods) and various microscopic organisms including cyanobacteria, algae, testate amoebozoans, various fungal groups, and oomycete fungal analogs (e.g., [4–16]). Associations between plants and fungi are a significant feature of the chert, and various fungi of the phylum *Mucoromycota* (i.e., the earliest mycorrhizal partners of land plants) and zoosporic fungi (i.e., fungi that reproduce with flagellated spores) have been described from this site (e.g., [4,12,14–23]). Many of these zoosporic fungi are parasitic, but to date, no unequivocal plant pathogen has been described. Of particular interest is *Paleopyrenomycites devonicus*, another representative from the Rhynie Chert, which was described as an ascomycete with sexual and asexual morphs colonizing the plant *Asteroxylon mackiei*[22,23]. Its potential roles as saprotroph, parasite, and/or pathogen have all been considered, but no firm conclusion has been reached[23]. Observations of the relationships of the hyphae with the host tissues away from the sporulating structures are lacking leaving its trophic strategy unresolved.

[1]Science Group, The Natural History Museum, London, UK. [2]Institut Systématique Évolution Biodiversité (UMR 7205), Muséum national d'Histoire naturelle, CNRS, Sorbonne Université, Paris, France. [3]Imaging and Analysis Centre, The Natural History Museum, London, UK. [4]Center of New Technologies, University of Warsaw, Warsaw, Poland. [5]Department of Earth Science & Engineering, Imperial College London, London, UK. [6]Department of Botany and Plant Pathology, Purdue University, West Lafayette, IN, USA. [7]Jodrell Laboratory, Royal Botanic Gardens, Kew, Richmond, UK. [8]Jilin Agricultural University, Changchun 130118 Jilin Province, China. [9]Geography and Environmental Science, University of Southampton, Southampton, UK. ✉e-mail: c.strullu-derrien@nhm.ac.uk

The early fossil record of ascomycetes is poorly documented and controversial. Evidence of hyphae and spores that could potentially represent ascomycete fungi date back to the Silurian[24]. Lichenized fungi, a group of symbiotic mutualists mainly represented by ascomycetes, have been reported from Paleozoic and Precambrian sediments. However, some researchers reject many of these early records for lack of compelling evidence[25,26]. A clear description of the characters of the fungi involved in these associations will help to confirm their affinity. The earliest generally accepted fossil lichens are the Early Devonian *Chlorolichenomycites salopensis* and *Cyanolichenomycites devonicus*[4,27]. Dispersed spores and other evidence of a major *Ascomycota* radiation do not appear in the sedimentary record until much later in the Mesozoic[26]. In this context, the species *Paleopyrenomycites devonicus* from the Rhynie Chert is of special relevance given that it is widely accepted as the earliest ascomycete and it is commonly used in the calibration of molecular phylogenetic trees of *Fungi*. However, the morphological characters of this fossil are challenging to interpret and *Pa. devonicus* does not fit neatly into any extant clade of *Ascomycota*.

Recently, an affinity with the *Ascomycota* has been suggested for *Prototaxites taitii*, based on the observations that it combines features of extant *Taphrinomycotina* and *Pezizomycotina*[28]. This combination of characters has not been observed in any living species, and the broad hymenial layer bearing polysporus asci, the character used to attribute it to the subphylum *Taphrinomycotina*, is not in organic continuity with the surface of the main axis of the type specimen, for which doubt exits about affinity. The hymenial layer only can be considered as belonging to *Ascomycota*. Moreover, large specimens of other *Prototaxites* species have also been interpreted as sporophores of a basidiomycete[29] or as a lichen with coccoid chlorophyte phycobionts and the fungus as belonging to *Mucromycotina* or *Glomeromycota*[30]. Consequently, the affinity of *Prototaxites* remains unresolved. While possibly fungal, it is so unlike any living taxon[26] that it may represent part of an extinct lineage lacking extant descendants[31].

Here we describe *Potteromyces asteroxylicola*, a new filamentous fungus likely related to the *Ascomycota* forming asexual spores (conidia) and colonizing the outer cortex and epidermis of axes and leaf-like outgrowths (enations) of the herbaceous lycopsid *Asteroxylon mackiei*. Our fungus has some features in common with the asexual morph of *Paleopyrenomycites devonicus* and provides an opportunity to review the affinity of that fungus and to discuss the reliability of using *Po. asteroxylicola* and *Pa. devonicus* for dating phylogenies in *Ascomycota*. The reaction tissues developed by the plant in response to *Po. asteroxylicola* are also novel and demonstrate that infection occurred before the death of the host. A pathogen is a disease-causing parasite. Our finding represents clear evidence of a plant reaction due to a fungal interaction causing pathogenicity. It further extends the already diverse range of fungal-plant interactions known to have taken place in early terrestrial ecosystems.

## Results

We report fungal colonization of aerial axes and enations of the plant *Asteroxylon mackiei*. The original description of the plant by Kidston & Lang[32] has subsequently been improved in terms of its spore-bearing structures[33] and in the anatomy and development of its rooting system[3]. The so-called enations are not considered to be true leaves since, unlike modern lycopsids, they do not develop internal vascular tissue. The fungus is represented by conidiophores arising in tufts (Figs. 1a–e, 2a–c, 3a–d, 4a, b, Supplementary Movies 1 and 2), which burst through the cuticle (Figs. 1c, d, 2c, d, 3a–d, 4a, b, Supplementary Movie 2); it causes dome-shaped surface projections (Figs. 1a–d, 2c, d, 3b–d, 4a, b, Supplementary Movie 2). Beneath the cuticle, the fungus develops a network of hyphae forming the stroma (Figs. 1c, d, 2c, d, 3d, 4a, Supplementary Movie 2). When the dome-shaped surface projection is well developed, fungal filaments are visible amongst plant cells that likely proliferate in response to the fungal colonization (Figs. 1d,

2c, d, 3d, 4a, Supplementary Movie 2). Conidia clearly develop at the top of the filaments (Figs. 1g, 2b, 3e, f, 4b, Supplementary Movie 1) and they appear to form separately (Figs. 1g, 2b, 3f, 4b, Supplementary Movie 1).

Based on the examination of our images, it is apparent that the tissue systems of the *Asteroxylon* plant had undergone some decay prior to fossilization. Cells of the xylem, the most distinctive and robust tissue system, are preserved as are the inner cortical tissues. Tissues of the outer cortex and epidermis are much less well-preserved, with an evident breakdown of cell walls. Remains of the branchiopod crustacean *Lepidocaris rhyniensis*, visible in each slide (Figs. 1a, 3b), indicate a wet environment, which might have hastened decomposition. However, it seems very unlikely that the fungal infection was post mortem. We have not found any report of a similar fungus from deposits of this age and therefore describe it as new here.

Taxonomy
Kingdom: *Fungi* R.T. Moore, 1980[34]
Subkingdom: *Dikarya* Hibbett et al., 2007[35]
Phylum: *Ascomycota* Caval.-Sm., 1998[36]
Class: *incertae sedis*
Genus: *Potteromyces* Strullu-Derrien & D. Hawksw., gen. nov.

*Etymology*: In honor of Helen Beatrix Potter (1866–1943) the well-known children's author, conservationist, and amateur naturalist, who used her artistic abilities to draw and document a variety of *Fungi*. She made detailed observations and was one of the first mycologists to study the growth of fungi from spores in culture and to understand that lichens were an association between an alga and a fungus[37,38]. She also had a keen interest in fossils[39]. She was a critical observer of fungi microscopically making novel and at the time controversial observations. Her contribution really merits acknowledgment in the fungal kingdom.

*Diagnosis*: Differs from other fungi known from the Rhynie Chert in having conidiophores arising in clusters from a loose sporodochium-like stroma located outside the cuticle. Conidia develop separately at the tip of the conidiophores, rounded, tending to form chains; conidiogenesis obscure, appearing annellidic or philaidic, the first-formed conidium holoblastic and later ones enteroblastic.

Species: *Potteromyces asteroxylicola* Strullu-Derrien & D. Hawksw., sp. nov. (Figs. 1–4).
*Etymology:* The specific epithet refers to the plant host.
*Diagnosis*: Conidiophores *ca.* 35, each 150–300 μm long and 7.5–11.5 μm wide, conidia 6.5–10.5 μm diam.
Holotype: *hic designatus*, NHMUK-V16430, thin section prep. c. 1915, W. Hemingway.
Locality: UK: Scotland, Aberdeenshire: Rhynie, fossil in Early Devonian chert (407.1 ± 2.2 Ma[40]).
Other material: Loc. cit., NMS G.1925-9-11 – isotype (thin section).
This published work and the nomenclatural acts it contains have been registered in Index Fungorum. The Index Fungorum numbers for this publication are IF 901132 (genus) and IF 901133 (species).

*Description*: Fungus colonizing the narrow outer cortex of aerial axes and especially enations of *Asteroxylon mackiei* (Figs. 1a, b, 3a, b). The hyphae form a sprorodochium-like stroma, 80–160 μm high from which the conidiophores arise (Figs. 1b–d, 2c, 3b, 4b). The epidermis of the plant is not preserved. When observed in a particular focal plane, the structure of the sporodochium is not clearly visible (Figs. 1b, c, 3b, c), but in different focal planes in normal transmitted light microscopy and especially in confocal microscopy it appears as formed of aggregate simple hyphae aligned vertically (Figs. 1d, 2c, 3b–d, 4a, Supplementary Movie 2). Conidiophores 7.5–11.5 μm wide and probably exceeding 150–300 μm in length emerge in tufts through the cuticle (Figs. 1c, d, 2c,

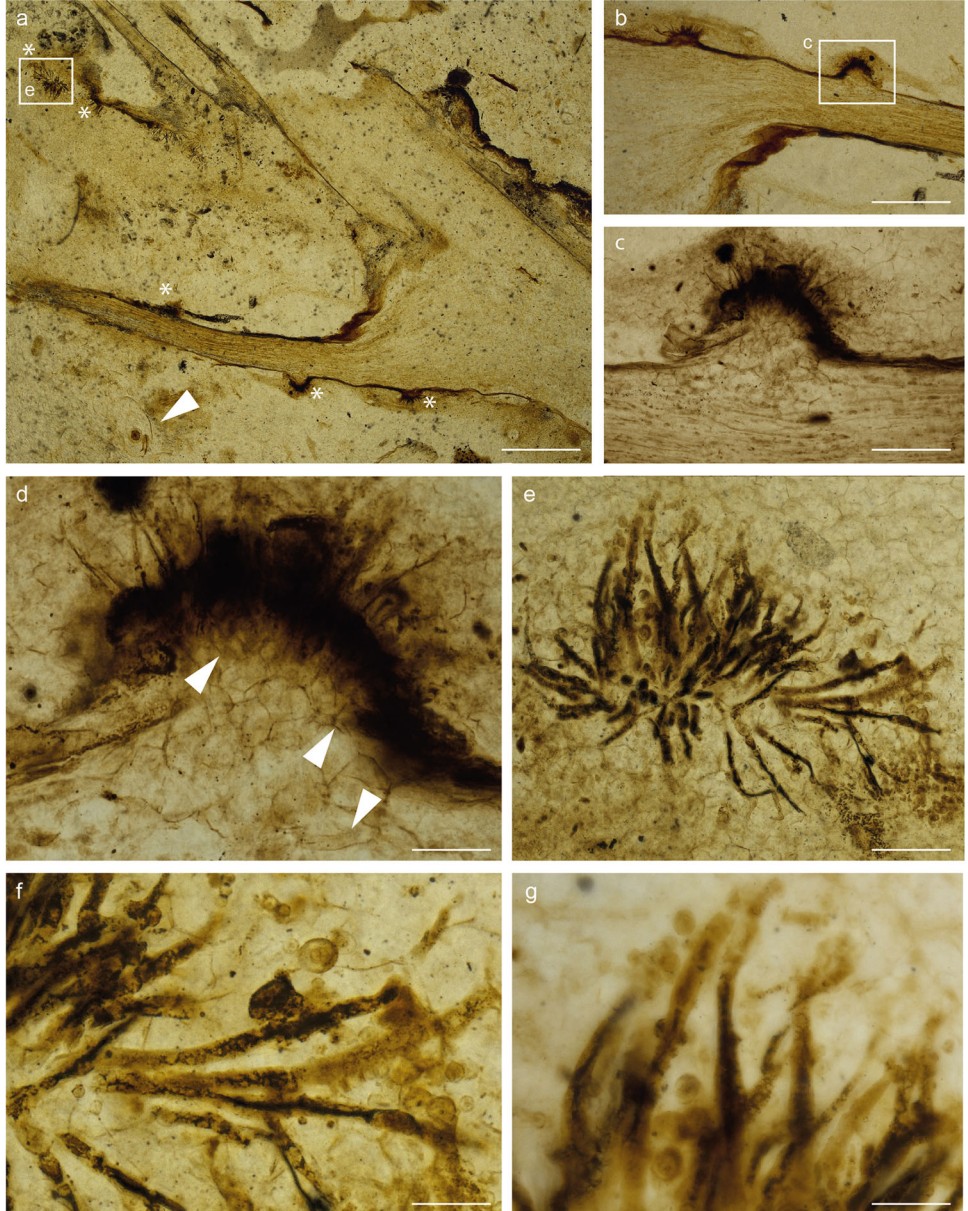

**Fig. 1 | *Potteromyces asteroxylicola* gen. et sp. nov. Holotype on thin section NHMUK-V16430 illustrated in white light. a** Location of the fungus (asterisks) along the aerial axis and enations of the plant *Asteroxylon mackiei*. Arrowhead showing animal remains. **b** Plant reaction induced by the fungal attack. **c, d** Higher magnifications of the framed zone from **b** showing the plant cell proliferation in response to the fungal attack. Arrowheads showing hyphae. **e** Conidiophores in tuft. **f** Higher magnification of **e**. **g** Conidiophores in high magnification, slightly shrunken at their distal end, and conidia. Scale bars: 900 μm (**a**); 400 μm (**b**); 120 μm (**c**), 45 μm (**d**), 85 μm (**e**), 35 μm (**f, g**).

d, 3c, d, 4a, b, Supplementary Movie 2), they are slightly shrunken at their distal end (Figs. 1e, g, 2a, c, 3e, f, 4b). Conidia develop at the top of the conidiophores (Figs. 1g, 2a–c, 3e, f, 4b, Supplementary Movie 1), arising separately, sometimes adhering in chains, with internal contents; they are rounded, more-or-less rough-walled, 6.5–10.5 μm in diameter. Conidia that appear along the sides of the conidiophores or at their bases (Figs. 1g, 2c, 3c) are evidently conidia that have fallen off the conidiophores. There is no direct link between these fallen spores and the conidiophores. Conidiogenesis appears annellidic or philaidic; the first conidium forms holoblastically (Figs. 1g, 2b, 3e, f, 4b; Supplementary Movie 1) and later ones form enteroblastically at least in some instances (Figs. 3f, 4b); it does not leave any visible scars. Plant cells within the main body of the enation are conspicuously elongate, whereas in the dome-shaped projections that arise as a reaction to the fungus they are more isodiametric and variable in shape and size

(Figs. 1d, 2c, d, Supplementary Movies 1, 2). Hyphae measuring 7.5–9 μm in diameter are visible among these cells.

## Discussion
### Similar fossil fungi
Taylor et al.[22,23] reported structures that they interpreted as an asexual morph of *Paleopyrenomycites devonicus* from the Rhynie Chert. This conclusion is circumstantial, being based on the asexual morphs being intermixed with ascomata assigned to the same species, in one case in proximity in the same section (Taylor et al.[23]; Fig. 15), but without evidence of any physical connection. The assumption that the sexual and asexual structures reported would have belonged to the same fungus and developed simultaneously remains questionable. Nevertheless, it is important to compare these asexual structures with the ones observed in the new fungus. In *Pa. devonicus*, they were described

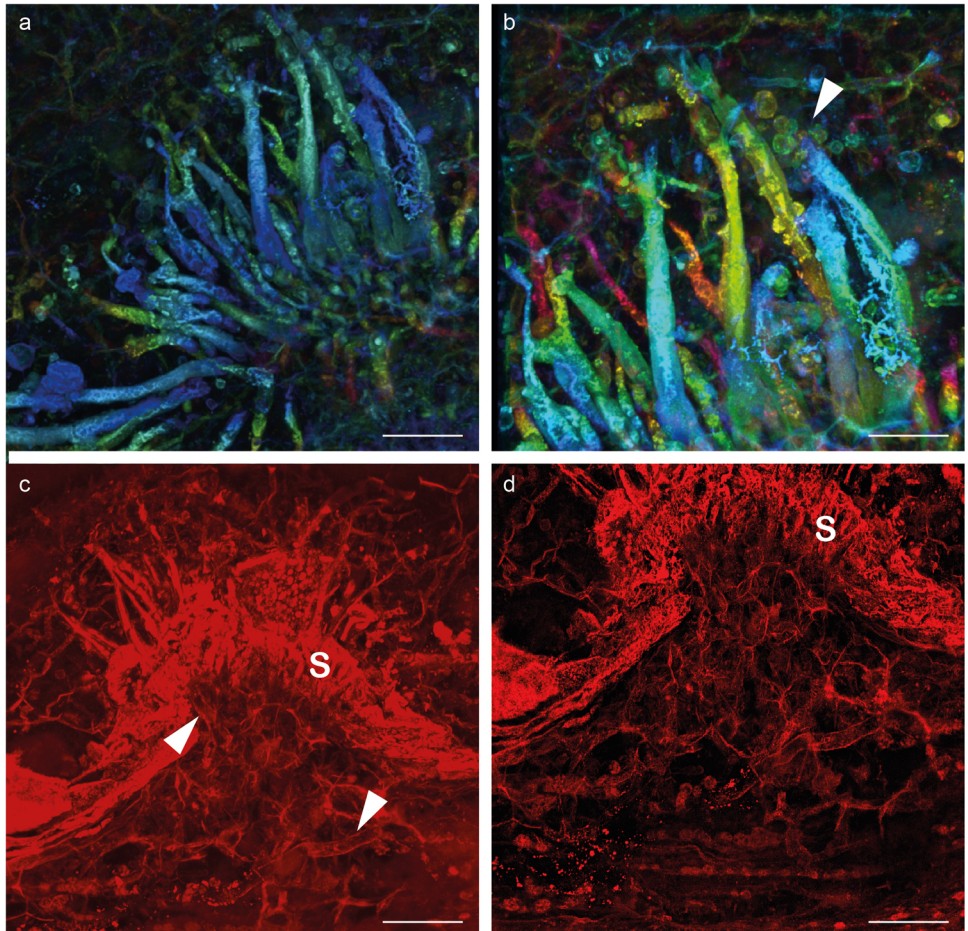

**Fig. 2 | Confocal laser scanning maximum-intensity projection images of** *Potteromyces asteroxylicola* **gen. et sp. nov. on thin section NHMUK-V16430.** **a** Same specimen as Fig. 1e. **b** Higher magnification of **a** showing conidiophores and conidia forming at the top of the conidiophore (arrow). False-colored for z-stack depth (**a**, **b**). **c** Same specimen as Fig. 1d showing hyphae (arrowheads) within the dome shape projection. **d** Same specimen as Fig. 1d; within the dome shape projection the plant cells are more variable in shape and size compared to the elongate plant cells devoid of fungus. S sporodochium-like stroma. Scale bars: 50 μm (**a**), 35 μm (**b**), 60 μm (**c**), (40 μm (**d**). See also the original datasets and animation movies: 10.5281/zenodo.7589422.

as acervuli, but a close examination of the published figures (Taylor et al.[23]: Fig. 36) revealed that they were not formed in a disc-like conidioma immersed in the host tissues, the characteristics of an acervulus (Kirk et al.[41]: Fig. 12 N), but rather on a sporodochium-like mound recalling the stroma-like structure seen in *Po. asteroxylicola*. Taylor et al.[23] described *Paleopyrenomycites* conidial development as "thallic, appearing to be holoarthric and basipetal" but also noted it might be "enteroblastic phialidic like that in the modern fungus *Chalara*" due to the suggestion of an outer wall around the chain of cuboid conidia (Taylor et al.[23]: Fig. 38, arrows), implying enteroarthric conidiogenesis in which the conidia arise by internal segmentation and then separation. There is also no unequivocal evidence presented to show that the two methods of conidiogenesis that they found and illustrated in different figures represent a single morph or species.

The conidiogenesis in *Po. asteroxylicola* is distinct from both of the asexual structures reported in Taylor et al.'s material as the first conidia are evidently formed holoblastically and singly at the tip of the conidiogenous cell (i.e., the segment of the conidiophore that produces the conidia). The morphology of the conidiophores also differs: in *Paleopyrenomycites* they are septate, up to 600 μm long and 10 μm wide; in *Potteromyces* they are aseptate and only 150–300 μm long and 7.5–11.5 μm wide. Another difference that could be significant is the location of the fungus on the host plant. *Potteromyces* colonizes both the aerial axes and the enations of *Asteroxylon*, whereas the alleged asexual morph of *Paleopyrenomycites* is considered to arise amongst the ascomata only on the aerial axes. Furthermore, the conidia of *Paleopyrenomycites* are nearly cuboid and 4-5 μm in diameter, whereas those of *Potteromyces* are rounded and 6.5–10.5 μm in diameter.

After examination of the available evidence on fossil fungi (e.g.[4,42]), we have failed to find any report of a fungus identical to that described as new here. There is some superficial similarity to *Fungites macrochaetus* described from much younger Paleogene (34–43 Mya) deposits, which has tufts of conidiophores, but they do not appear to be compacted and there is no evidence of a basal stroma[43,44]. Kettunen et al.[45] reinvestigated some of the specimens from Caspary & Klebs[42,43]. In their description of the holotype of *Fungites macrochaetus*, they reported that straight or flexuous conidiophores arise singly or in tufts or loose fascicles from the surface of a small, stalked, angiosperm fruit. These conidiophores appear to be longer [(90) 300–500 (600) μm long] and with a rather larger diameter [5–14 μm wide (3–8 μm at the tip)], than the conidiophores of *Po. asteroxylicola*. Caspary & Klebs[43] mentioned that no spores were visible in the conidiophores. However, Kettunen et al.[45] were able to detect numerous minute conidia, both attached to the conidiophores and free in the surrounding amber matrix. The conidiogenous cells are nodelike, polyblastic, at first terminal, but later also intercalary as the conidiophore grows in height. Conidia are simple, ellipsoidal to ovoid, minute (1–2.5 μm wide, 3–5 μm long) and they do not appear to have been produced in chains, or at least no chains have been observed. Therefore, conidiogenesis and conidia in *Fungites macrochaetus* differ from those observed in our fungus.

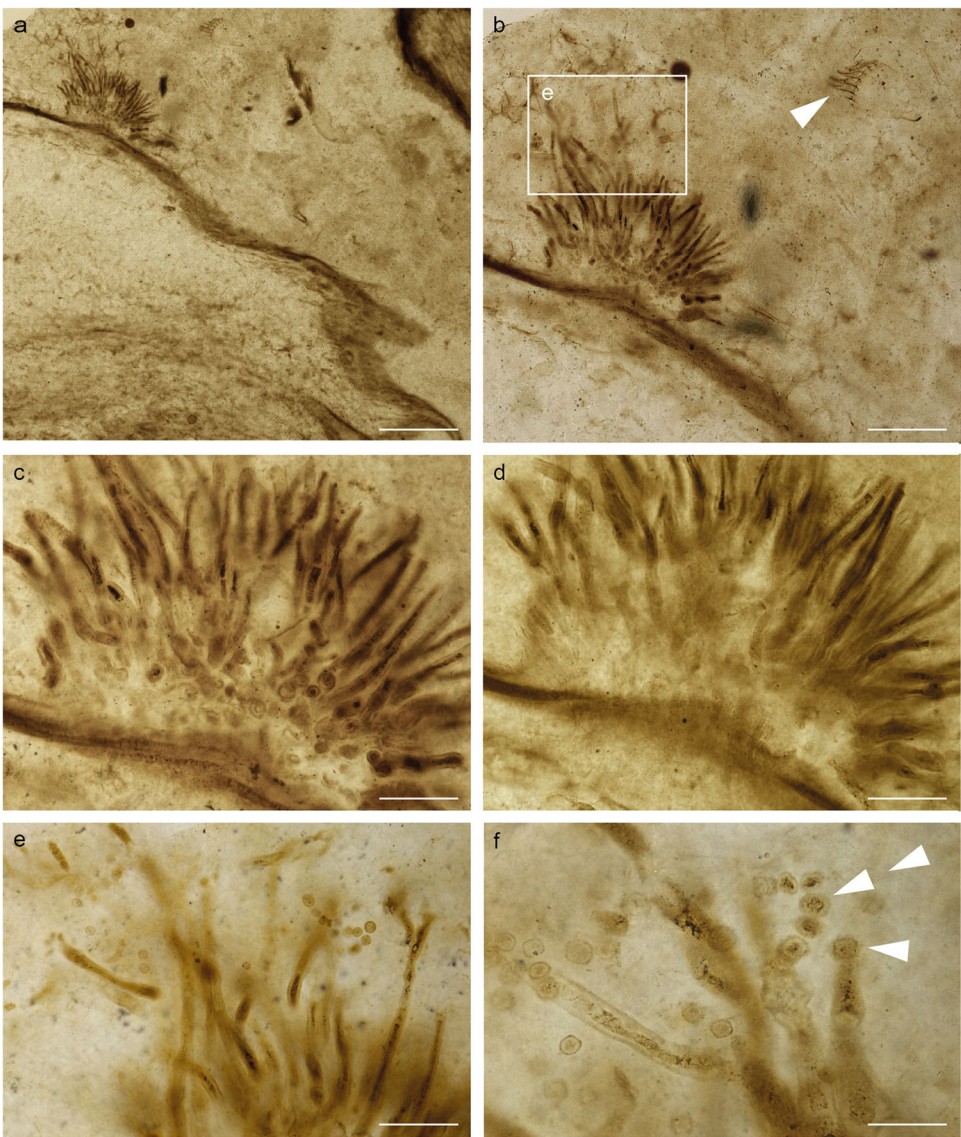

**Fig. 3 | Isotype of *Potteromyces asteroxylicola* gen. et sp. nov. on thin section NMS G.1925-9-11 illustrated in white light. a** Tuft of conidiophores arising from beneath the cuticle of an aerial axis of the plant *Asteroxylon mackiei*. **b** Higher magnification of **a**. Arrowhead showing animal remains. **c**, **d** Higher magnifications of **b** in different focal planes. Sporodochium-like stroma made of vertically aligned hyphae visible in **d**. **e** Higher magnification of the framed zone from **b**. **f** Higher magnification of **e** showing conidiogenesis; first-formed conidium holoblastic (arrowhead); conidia in chain (double arrowheads). Scale bars: 300 µm (**a**), 125 µm (**b**), 55 µm (**c**, **d**); 45 µm (**e**); 30 µm (**f**).

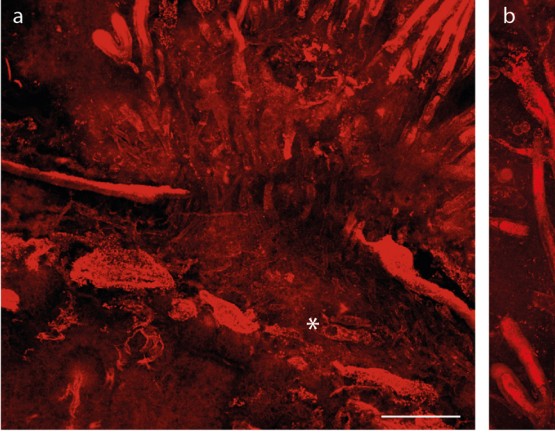
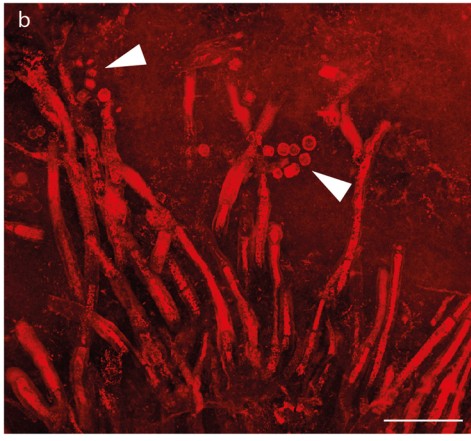

**Fig. 4 | Confocal laser scanning maximum-intensity projection images of *Potteromyces asteroxylicola* gen. et sp. nov. on thin section NMS G.1925-9-11. a** The fungus bursts through the cuticle of an aerial axis of the plant *Asteroxylon mackiei*; asterisk showing a cut hypha. **b** Tuft of conidiophores arising from the cuticle. Conidia in chains (arrows). Scale bars: 50 µm. See also the original datasets and animation movies: 10.5281/zenodo.7589422.

## Similar extant fungi

Neither the organization nor the morphology of the asexual morphs of *Paleopyrenomycites* and *Potteromyces* provide unequivocal evidence of close relationship to any extant taxonomic groups within the *Ascomycota*. There are, however, numerous extant fungi that grow on plants and which have hyphae immersed in leaves that produce sporodochium-like tufts of conidiophores erumpent through stomata or epidermal layers. These belong to diverse taxonomic groups, but, because of their similar habit, with asexual spores produced without any enclosing structure, they have been collectively conveniently referred to as "hyphomycetes". In the past, asexual and sexual morphs of the same species, so-called pleomorphic fungi, were allowed to have separate scientific names, but that practice was discontinued in 2011, so now each species has just one name whether it is known only as an asexually sporing fungus, only as a sexually sporing fungus, or has both asexual and sexual sporing morphs[46]. Information on formally described genera of hyphomycetes, including illustrations of over 1,400 genera is provided in Seifert et al.[47]. The most similar structures, with conidiophores arising from a loose stroma and bursting through the leaf surface, are seen in several so-called cercosporoid genera (*op. cit.*: Plates 254-6), some of which produce conidia enteroblastically from pores leaving scars (e.g., *Cercospora*), while others produce them in a holoblastic manner (e.g., *Pseudocercospora*) not dissimilar from that seen in *Potteromyces*, but producing multiseptate rather than simple globose conidia.

Within the *Basidiomycota*, only the spermatia produced within spermogonia (one of five spore stages) of rust fungi (*Pucciniales*) share the character of phialidic spore formation[48,49] with *Potteromyces*. Where spermatia formation has been studied, spores are formed by basipetal succession from the apex of the spermatiophore and then break off to fill the spermogonial cavity. However, in marked contrast to *Potteromyces*, the spermogonia of extant rust fungi are ± flask-shaped; those of the earliest extant rusts, i.e., *Rogerpetersonia* and *Mikronegeria*, are very deep seated within the host tissue with only the flask necks protruding[50,51]. Additionally, the rust fungi themselves are a more recently diverging lineage of *Pucciniomycotina*, estimated to have evolved c. 200 Mya[50]; spermogonia and phialidic spore production is not known from any of the sister lineages to *Pucciniales*, or elsewhere within *Pucciniomycotina*[52], making it likely that phialidic sporogenesis evolved independently within *Pucciniales* from *Ascomycota*. Finally, *Pucciniales* are biotrophic plant pathogens, whereas *Potteromyces* appears to be necrotrophic, and in extant rust species that form spermogonia, these are accompanied by the formation of a second sorus type, termed an aecium and characterized by production of catenulate chains of spores within a cup-shaped sorus that originates from and is formed in close proximity to the dikaryotinized hyphae of the spermogonium. The lack of any analogous structure in *Potteromyces* also makes it very unlikely that it represents a species of *Pucciniales*.

Based on the information gleaned from our study of this novel genus, it seems most appropriate to regard it as of uncertain position, but most likely as an ascomycete *incertae sedis* belonging to an extinct clade.

## Lifestyle of *Potteromyces* and the evolution of pathogenicity

Despite the lack of well-documented *Dikarya*, the fungi of the Rhynie Chert geothermal system were both taxonomically diverse and varied in their ecological roles. A zoosporic fungus is known as a parasite of the Rhynie Chert charophyte *Palaeonitella cranii* where it induces cell hypertrophy in the host[12]. A few other examples of zoosporic fungi that were possibly parasitic have been documented in the axes and spores of Rhynie plants. However, establishing proof of a parasitic relationship between the fungus and its plant host is challenging. One operculate fungus has been observed on the cellular gametophyte emerging from the proximal surface of an *Aglaophyton* spore[15]. The authors reported that a parasitic interaction might exist, however, in the absence of a recordable host response they concluded that this was difficult to substantiate. *Palaeozoosporites renaultii* is another fungus that colonized the inner cortex of the rooting system of *Asteroxylon mackiei*, but the nature of its interaction with the plant remains unclear. The distinctive restricted distribution of the fungus, to the inner cortex of the rhizome where the cells are degraded, was thought to be more suggestive of a parasitic relationship[19]. Three fungi were observed in the rhizome and rhizoids of *Nothia aphylla*[4,20,21]. These fungal endophytes were associated with cell and tissue alterations and host responses; the authors considered that the associations were either specific (i.e., association with only one of the fungi) or non-specific (i.e. association with one fungus or association with two or all three fungi). One of them was described as mycorrhizal, which is symbiotic. The second one was described as responsible for secondary cell wall thickenings, however, this represents a harmless reaction from the plant against a fungus that was suggested as possibly chytrid-like. The third one was described as a parasite because two of the infected rhizoidal cell ridges of the plant were reported to contain peripheral regions that were devoid of cells. By analogy to some extant plants, it was suggested that this tissue degradation might have been the result of a defense mechanism against the fungus. In general, phytopathogenic microorganisms are deterred by programmed cell death around the infected areas, which inhibit the microbes from spreading. However, in this case colonization by the fossil fungus was not clearly demonstrated around that zone (Fig. 3b Krings et al.[21]; Fig. 4.56 and Fig. 4.57, Taylor et al.[4]). Regardless of their hypothetical parasitic roles, none of these examples show definite features of pathogenicity.

*Potteromyces* colonized the plant *Asteroxylon mackiei*, which has been found in association with the branchiopod crustacean *Lepidocaris rhyniensis* (Figs. 1b, 3a). The plant grew on a sandy surface close to shallow pools[53]. The association with *Lepidocaris rhyniensis* likely happened post mortem when the plant was washed into the pool or when the plant was submerged during a flooding event. The fossilized remains of the plant provide clues to the life history strategy of *Potteromyces*. The fungus evidently bursts through the cuticle provoking damage to the plant resulting in reaction tissues directly associated with the fungus, leading to the development of distinctive dome-shaped projections arising through proliferation of epidermal or subepidermal cells (Figs. 1a–d, 2e, f, 3b–d, 4a, b). The cells within these projections are smaller and more varied in shape than the regular elongate cells of the neighboring tissues of the plant axis (Figs. 1c, d, 2f). Reaction tissues have been reported in the Rhynie Chert (e.g., [53–55]), however, these developed in cells neighboring large cavities within axes. The organisms responsible remain unknown, but arthropods and nematodes have been implicated. Reactions of the type here described, forming in response to a fungus, have not been observed in Rhynie Chert plants and they demonstrate that infection occurred before the death of the host. This provides strong evidence that *Potteromyces* was a pathogen of *Asteroxylon*, and the outward manifestation of disease was likely to have been necrotic patches corresponding to the dome-shaped projections on the leaf-like enations and axes of the plant. Inducing host necrosis is a strategy of necrotrophic plant pathogens which first kill host tissues and then derive their nutrition from the dead host cells. This finding is early evidence of a plant-fungal pathogen in the fossil record, and it extends the already diverse range of fungal-plant interactions known to take place in early terrestrial ecosystems.

Extant fungal plant pathogens are found throughout the phylum *Ascomycota* and primarily in two subphyla of *Basidiomycota*. In the ascomycetes, they occur in classes such as *Dothideomycetes*, *Leotiomycetes*, or *Sordariomycetes*. In the basidiomycetes, most of the plant pathogens are rusts and smuts (*Pucciniales* and *Usilaginomycotina*, respectively)[56]. Plant pathogenic biotrophic ascomycetes were reconstructed as having evolved more recently than biotrophic basidiomycetes[57]. However, James et al.'s[57] phylogeny did not separate biotrophs from other kinds of pathogens such as necrotrophs. Also,

according to Ohm et al.[58], the ancestor of *Dothideomycetes* may have been either a saprotroph or a plant pathogen. The early fossil record of *Basidiomycota* is weaker than that of *Ascomycota*. Wood decay has been reported as a possible early example of white rot involving *Basidiomycota* by the Late Devonian (c. 372–359 Mya)[59]. However, neither clamp connections nor any basidiome were observed, leaving open the possibility that another type of fungus was responsible. Later evidence of *Basidiomycota* comes from hyphae with clamp connections towards the end of the Carboniferous (c. 330–305 Mya)[60]. Krings et al.[60] suggested that if the callosities formed in response to invading clamp-bearing hyphae, then this host response would favor evidence of a parasitic infection; they wrote that the fungus could have been biotrophic or saprotrophic, but did not mention the possibility of pathogenicity. Our study shows that early fossil evidence for pathogens comes from an ascomycete.

### Dating the phylogenetic tree of *Ascomycota*

*Paleopyrenomycites devonicus*[22] was first attributed to *Ascomycota* based on the presence of an ascoma (perithecium) containing asci, the synapomorphy for the phylum. However, the cladistic analysis performed by Taylor et al.[23] did not resolve its relationship to other groups of fungi. Furthermore, certain key traits are problematic in various ways[61] and cannot confidently be referred to homologous structures. For example, the operculum of the ascus is not clear in the original illustrations. Among extant *Ascomycota*, operculate asci only occur in apothecial ascomata (i.e., open and disc-like, not covered and with an opening)[62], whereas in this fossil they are associated with perithecial ascomata (i.e., rounded to flask-shaped with a pore-like opening, an ostiole). Perithecial ascomata facilitate forcible discharge of the ascospores[63], but the ascospore shape and orientation in *Pa. devonicus* are not consistent with forcible discharge. Other characters of *Pa. devonicus* are prone to homoplasy; the ostiole, a distinct apical pore in the ascoma, is known to have evolved independently at least four times in *Ascomycota*. The fossil also has a further suite of features that are not found together in any living species[61].

The exact placement of *Pa. devonicus* in the *Ascomycota* clade and its relationship to modern classes and orders within *Ascomycota* is still debatable. It has been widely used and it is still used in dating the fungal tree of life[26]. However, whether it is basal to the *Ascomycota* or nested within the subphylum *Pezizomycotina* is contentious and its position varies depending on the studies and data used leading to strikingly different age estimates for the same divergence events (e.g.,[26,64–69]). For example, Lücking et al.[66] considered using this fossil to calibrate three alternative positions in the *Ascomycota* tree: origin of *Sordariomycetes*, divergence of *Pezizomycotina*, and origin of *Pezizomycotina*. They recommended adoption of a conservative approach by using the fossil as a representative of the stem age of *Pezizomycotina*. This placement was subsequently followed by Nelsen et al.[64] and Prieto & Wedin[67] who performed a thorough review considering alternative fossil constraints and discussing the impact of various calibration scenarios on age estimates. Conversely, Lutzoni et al.[69] placed this fossil on a more recent node, the divergence of *Sordariomycetes*. Given that paleontological evidence seems to indicate an *Ascomycota* crown group radiation during the Mesozoic Era, Berbee et al.[26] argued that *Pa. devonicus* probably represents an extinct clade of early-diverging *Ascomycota*.

Although various analytical methods try to deal with and reduce the uncertainty associated with the phylogenetic placement of fossil calibration points, incorrect fossil ages (e.g.,[70–72]), and the use of fossils without key defining synapomorphies for the group[73], the reality is that an erroneously placed fossil, due to a wrong interpretation, may result in a cascade of misleading conclusions about the evolution of *Fungi* and their interactions with other organisms. These errors might have the greatest effects in deep nodes. So we caution against reliance on early fossils in dating evolutionary radiations of different ascomycete classes. In the case of *Potteromyces asteroxylicola*, while an ascomycete

crown group and even a *Dothideomycetes* affinity is possible, its phylogenetic position remains uncertain. Further details are needed, such as the ultrastructure of the hyphal walls. We consider that *Po. asteroxylicola* belongs to an extinct lineage of *Ascomycota* attributed to the *Ascomycota* stem group. It could be justified for use as a minimum node age calibration point for the whole *Ascomycota* (i.e., as the stem node of *Ascomycota* in a phylogeny) or perhaps even the *Dikarya* crown group. The same suggestions apply to *Paleopyrenomycites devonicus*[22,23] and to the hymenial layer bearing polysporus asci attributed to *Prototaxites taitii*[28].

## Methods

We examined and photographed specimens in two thin sections (i.e., paper-thin slices of rock glued to microscope slides): NMS G.1925-9-11 from the National Museum of Scotland, Edinburgh, and NHMUK-V16430 from the Natural History Museum, London. Both The National Museum of Scotland, Edinburgh, and the Natural History Museum, London, have approved the study protocol.

The thin sections were prepared early in the 20th century by professional geological slide maker W. Hemingway. The Rhynie Chert itself was discovered in 1912 by William Mackie (1856–1932), a medical practitioner surveying the regional geology. Palaeobotanists William H. Lang and Robert Kidston initially described five plants from these deposits, one of which was *Asteroxylon mackiei*[32].

These cherts are rocks, originally deposited as silica-rich sinter from ancient hot springs. The Rhynie Chert site is located near the village of Rhynie, (57° 18′ 60.00″ N, 02° 49′ 59.99″ W), situated about 50 km north-west of Aberdeen, Scotland. Plants grew on or close to a sinter surface in an open landscape that included shallow freshwater pools and sites of raised relief that were drier[74]. Radiometric age of the Rhynie Chert is 407.1 ± 2.2 Ma ($^{40}$Ar/$^{39}$Ar age)[40]– biostratigraphic age of Pragian-?earliest Emsian[75].

The specimens were studied with a Nikon Eclipse LV100ND compound microscope. Depth of field was enhanced through z-stack montage. Confocal images were obtained with a Nikon A1-Si confocal laser scanning microscope, and we used a 40x oil-immersion objective and selected an area of interest around the object for digital enlargement. An autofluorescence signal was collected using four photomultiplier detectors (425–475 nm for the 405 nm laser, 500–550 nm for the 488 nm laser, 570–620 nm for the 561 nm laser, and 675–725 nm for the 640 nm laser). Samples were visualized with a 30 μm confocal pinhole, and 100–400 optical slices per z-stack were acquired for each detector. The fluorescence signal from each z-stack was then projected onto a false-color maximum-intensity projection image and used to generate depth-coded visualizations of the sample (Nikon NIS-Elements software, http://www.nis-elements.com) (see[16,19] for additional information). The resultant z-stack datasets (.nd2) and animation movies (.mp4) have been uploaded to a Zenodo repository[76].

### Statistics and reproducibility

Images show microscopic observations, not experiments repeated independently. Reproducibility is through re-examination of the original physical specimens.

### Reporting summary

Further information on research design is available in the Nature Portfolio Reporting Summary linked to this article.

## Data availability

All confocal data collected and used in this study, plus animated videos showing the data and 3D models, are deposited in the Zenodo repository[76] under a Creative Commons Attribution 4.0 international license. https://doi.org/10.5281/zenodo.7589422. Thin section NHMUK-V16430 is housed at the Natural History Museum, London. Peta Hayes

(p.hayes@nhm.ac.uk) can be contacted for access. Thin section NMS G.1925-9-11 is housed at the National Museum of Scotland, Edinburgh, Andrew Ross (A.Ross@nms.ac.uk) can be contacted for access.

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

## Acknowledgements

The authors thank Andrew Ross (National Museum of Scotland, Edin-burgh) and Peta Hayes (Natural History Museum, London) for access to the specimens. They acknowledge Keith A Seifert, Mary Berbee, Sébastien Duplessis for helpful discussion about the fossil structures. CS-D thanks the Fondation ARS Cuttoli-Paul Appell/ Fondation de France for supporting her work on fossil fungi (grant no. 00103178) and a grant to EG (RBG Kew) by the Evolution Education Trust.

## Author contributions

C.S.-D. conceived the study. C.S.-D., T.G., A.R.T.S. and P.K. acquired specimen photographs and tomographic data. C.S.-D. and D.L.H. ana-lyzed the data with the help of E.G. and M.C.A. and wrote the first draft. All authors contributed to the final version of the manuscript.

## Competing interests

The authors declare no competing interests.
