## [Peer Review File · Nature Communications]

A fungal plant pathogen discovered in the Devonian Rhynie ChertReviewers' Comments:

Reviewer #1:

Remarks to the Author:

This manuscript describes a new plant-associated fungus in the Lower Devonian Rhynie Chert. Rhynie is of enormous importance as the oldest terrestrial ecosystem that is preserved with anywhere nears its amazingly complete detail. Any new discovery is worth celebrating and the discoveries keep coming. The authors of this study include some of the most active explorers of Rhynie in recent years and have been pivotal in the continued expansion of our understanding of this ecosystem. This new taxon is a worthy addition to that legacy. The question becomes what to do with it, and there are issues with the context in which the fossil has been placed.

The authors emphasize the significance of the find in two directions: 1., as an early (maybe?) ascomycete that may be useful for time calibration of the fungal phylogeny and, 2., as an oldest fungal pathogen. Neither of those arguments hit exactly right.

Point 1:

The paper ends with a section on time calibration of phylogenies where the conclusion is "Described as Ascomycota incertae sedis, *Potteromyces asteroxylicola* fits into the Ascomycota stem group and could be justified for use as a minimum node age calibration point for the whole Ascomycota (i.e., as the stem node of Ascomycota in a phylogeny) or perhaps even the Dikarya crown group." (Lines 424-426) This feels problematic on two levels. First, so many first appearances stack up at Rhynie, that only the most derived fossils end up mattering for time calibration. In the same section the authors discuss *Paleopyrenomycites* and whether it is stem *Pezizomycotina*, crown *Pezizomycotina*, or maybe even within *Pezizomycotina*, but all of those options are crown Ascomycota. So, the new fossil being stem-group Ascomycota makes it irrelevant for dating, since *Paleopyrenomycites* is there already in the same rocks and is more derived within Ascomycota. There is no calibration the new fossil can offer that *Paleopyrenomycites* can't provide already.

Second, there is nothing diagnostic about the new fossil to make it conclusively an ascomycete, and such a fossil simply should not be used for dating. The manuscript even offers mention of another fossil that can serve as a neutral example that is equivalent. The Late Devonian wood rot described by Stubblefield et al is of great significance for paleoecology but is of no value at all for time calibration because we don't know what that fungus is exactly. Most rots are basidiomycetes, but no diagnostic traits are available, so that rot fossil has been ignored when calibrating the basidiomycete phylogeny, and correctly so. The same logic applies to the new fossil described here and the ascomycete phylogeny. On multiple levels, value for the new fossil as time calibration really isn't there.

Point 2:

"Our finding also supports an affiliation to Ascomycota rather than to Basidiomycota for the origin of fungal plant pathogenicity." (Lines 51-52)

"these zoosporic fungi are parasitic, but to date no unequivocal plant pathogen has been described." (Line 75)

"Zoosporic fungi are common parasites of the Rhynie chert charophyte *Palaeonitella cranii* where they are known to induce cell hypertrophy in the host (Taylor et al. 1992b)." (Lines 314-315)

The distinction between parasitic versus pathogenic might not be obvious to much of your readership—it was not immediately obvious to me—so you should define it clearly from the outset. As eventually explained, the key distinction appears to be whether or not the plant shows a response, although nothing described here seems as pronounced as the reaction tissue used to verify that putative arthropod herbivory in fossil plants was not post mortem. But if the distinction being drawn is of a response from the host, then why does the cell distension mentioned for *Palaeonitella* not count? Is it a question of host cell enlargement versus host cell divisions? Is it because you're only after land plants? Wherever the line is being drawn, it feels like a needle is being thread.

Other things:

"Each enation is associated with a single vascular system" (Line 165)

That should be "associated with a single vein" -- the whole plant only has a single vascular system.

"Etymology: The generic name honours Helen Beatrix Potter (1866–1943)" (Line 192)

I was pleased to learn that Beatrix Potter was a mycologist. Thanks for that.

"Hence, the association with *Lepidocaris rhyniensis* probably happened post mortem when the plant was washed into a shallow pool." (Lines 342-344)

Or the water levels rose without needing the plant material to be transported?

"Extant fungal plant pathogens are mostly found in the phylum Ascomycota and to a very limited extent in Basidiomycota." (Lines 358-359)

This doesn't feel fair to Basidios. At 10,000 extant species, rusts and smuts represent a third of basidiomycete diversity. Plus, if you think about the basal topology of the basidiomycete phylogeny, they are one half of the tree.

Reviewer #2:

Remarks to the Author:

The authors describe a new fungus from the Early Devonian Rhynie chert that is attributed to the Ascomycota incertae sedis. It is said to be the earliest evidence of a plant pathogenic fungus. Yes, this work will be significant for related fields because it has an impact on our ideas on fungal plant pathogenicity and their origins. The work largely supports the conclusions and claims but some points need to be demonstrated in more detail, i.e. the aseptate nature of the conidiophores and the part on the formation of conidia. Some citations and data cited from the literature are incorrect and need to be corrected. Proper diagnoses should be given. The methods section is fine. Attached is an annotated version of the manuscript and a file with comments.

Hans Kerp

This is an interesting manuscript that is definitely worth to be published. However, several points still need attention (see below). Further remarks are made in the annotated pdf file. One of my concerns are the illustrations. They are of good quality, but there are quite a number of duplicates. I really wonder what the many confocal images really add to the excellent light-microscopic pictures. The latter are mostly clearer and suffice. Therefore, I suggest to reduce the number of confocal images to the absolute minimum and retain only those that aspects that are not to be seen or on the LM images

- I. 45 Why “the earliest **unequivocal** fossil evidence of a plant pathogenic fungus”, when the title reads: “The earliest fungal plant pathogen?”. The word “unequivocal” in the text and the question mark in the title seem to be conflicting.
- I. 74 Add: Krings, M. *Triskelia scotlandica*, an enigmatic Rhynie chert microfossil revisited. *PalZ* **95**, 1–15. (2021).
- I. 90 replace “early” by “Lower”. Early refers to age or time, whereas Lower refers to a rock series, as is the case here. The adjective Lower must be capitalized because the Devonian is formally subdivided into Lower (Early), Middle and Upper (Late) (see the latest version of the IUGS International Chronostratigraphic Chart). Therefore, it should be “...from the Lower Devonian”; Also correct would be: “The earliest generally accepted fossil lichens are the Early Devonian *Chlorolichenomyces salopensis* and *Cyanolichenomyces devonicus*”. See also correction in I. 367. Please check the use of stratigraphic nomenclature throughout the manuscript.
- L. 92 “>250 Mya” means more than 250 Myr, i.e. in the Paleozoic!
- I. 91 The correct orthography is *Prototaxites taitii* as it was named after Mr. D. Tait, fossil collector to the Geological Survey, who opened the first trenches in the Rhynie field in 1913.
For details on the orthography of epithets see Art. 60.8 of the International Code of Nomenclature for algae, fungi, and plants (Shenzhen Code).
- I. 91 Normally, in taxonomic literature only names of genera and subordinate taxonomic units are printed in italics, and names of higher rank (families, orders, classes, phyla) not. However, this may depend of the style of the journal.
- I. 105 Retallack & Landing (2014) interpreted *Prototaxites* as a belonging to the *Glomeromycota*.
Retallack G.J. & Landing, E. Affinities and architecture of Devonian trunks of *Prototaxites loganii*. *Mycologia*, **106**: 1143–1158. (2014)
- I. 115 Do not abbreviate genus name because this is very confusing, because in the same paragraph you refer to two genus names starting with *P*.
- I. 163 ff. It is correct that the enations of *Asteroxylon mackiei* lack vascular tissue. However, the sporophylls are vascularized. It is the question whether the lack of internal vascular tissue in the enations is a primary condition, or whether it is a secondary loss.

- I. 178 See comments to I. 339-344 below.
- I. 188 I have some concern with regard to the correctness of the name and the orthography of the epithet. According to Art. 62.2(a) of the Art. 60.8 of the International Code of Nomenclature for algae, fungi, and plants (Shenzhen Code), compounds ending in *-myces* are masculine, irrespective after whom the genus is named. The inflection of the epithet should be accordingly. The correct name would thus be: *Potteromyces asteroxylicolum*.

Shenzhen Code:

62.2. Compound generic names take the gender of the last word in the nominative case in the compound (but see Art. 14.11). If the termination is altered, however, the gender is altered accordingly.

(a) Compounds ending in *-botrys*, *-codon*, *-myces*, *-odon*, *-panax*, *-pogon*, *-stemon*, and other masculine words, are masculine.

- L. 199 ff. I have serious problems with the diagnosis, which is in my opinion not a proper diagnosis. A diagnosis should be a brief characterization of the taxon, highlighting the features in which it differs from closely related and/or similar taxa. Exact measurements do not belong in a diagnosis but in the description. Here measurements are given with a precision of 0.5 μm . What to do with specimens that are morphologically 100% identical but just 0.5 μm narrower or wider? These do not fit the diagnosis! Moreover, when taxa are first described usually little is known about the natural variability, especially in fossil material. Moreover, a combined diagnosis for the genus and species is given. According to the latest edition of the nomenclature code which supersedes all previous editions it is possible to give a combined generic/specific diagnosis. However, it should be noted that such diagnoses are often so tightly defined that it is hardly possible to include additional species. Therefore, I personally prefer separate diagnoses, the generic one being a bit more general than the specific one. Last but not least, according to the current nomenclature rules, it is not strictly necessary to include a diagnosis; a good description will suffice. Nevertheless, I prefer including diagnoses.
- L. 222-4 Couldn't this be a matter of preservation?
- I. 253-5 This statement should be further substantiated, because it is not really convincing. Please provide pictures with accompanied explanatory drawings to make this clearer.
- I. 257 Are the conidiophores in *Potteromyces* really aseptate? Figs. 2C and 2D seem to show septa.
- I. 266 Please note that the term Tertiary is no longer used. It has been replaced by Paleogene and Neogene (see International Chronostratigraphic Chart). Moreover, the ages given here are for Caspary's material incorrect. Kettunen *et al.* (2019. p. 367) wrote:
"These amber-bearing strata are Priabonian (c. 34–38 Ma) in age, but small amounts of amber also occur in Lutetian and Oligocene sediments, leading to a possible age range of c.25–43 million years for all strata bearing Baltic amber (Kosmowska-Ceranowicz et al. 1997; Standke 1998;

Kasínski & Kramarska 2008; Standke 2008). It is unclear whether the Oligocene amber represents redeposited Eocene material (Standke 2008); thus, a Lutetian to Priabonian age (c. 34–43 Ma) of Baltic amber is currently considered. Baltic amber eroded from these sediments is often found washed ashore along the coast of the Baltic Sea (predominantly in the Baltic States, Poland, Denmark, Germany and in southern Sweden) and in adjacent areas, and a large proportion of historic and new amber collections contains this ‘sea amber’. A precise locality of origin therefore can-not be provided for Baltic amber pieces from historic collections that were developed in the Königsberg (Kaliningrad) and Danzig (Gdansk) areas during the nineteenth and early twentieth centuries. This fact does not, however, affect the age estimate given above since the vast majority was initially embedded in these Eocene sediments (Standke 2008).”

I. 314 Zoosporic fungi are said to be common parasites in the Rhynie chert. How common?

I. 328 ff. Sorry, but here I fundamentally disagree! The secondary cell wall thickenings are clearly a response to one particular fungus.

I. 339-44 These phrases are a bit confusing. *Asteroxylon mackiei* did not grow in wet environments as the presence of stomata on root-bearing axes demonstrates. *Asteroxylon mackiei* is commonly associated by *Horneophyton lignieri*. They grew on sandy, probably well-drained soils. *Asteroxylon* had roots that penetrated the soil. In my opinion, there is no doubt that the association with *Lepidocaris* is post mortem. Probably the plant was washed into a pool. However, it should be noted that *Lepidocaris* fossils are usually exuviae, which are very light and transported very easily, even by wind from ephemeral pools. I suggest to shorten this part to make it clearer.

I. 350 This is a rather bold statement. Such reaction tissues being evidence for regeneration by Rhynie chert plants have been illustrated before, e.g., Kidston & Lang (1921a), Edwards & Selden (1992), Krings (2021).

I. 372 Callosities have associated with the clamp-bearing hyphae been described by Krings *et al.* (2011, p. 20)

I. 464 ff. The references to Caspary are incorrect, respectively incomplete. Both works were published posthumously. The text volume was published in **1906** (not 1907!) and the accompanying atlas in 1907. Robert Caspary was a German botanist (1818–1887), who was born in Königsberg (now Kaliningrad, Russia). After stays in Bonn, Berlin, Italy, England and France he was appointed professor in botany at the University of Königsberg in 1859. The so-called casparian strips are named after him.

His work on Baltic amber was published some 20 years after his death, in 1906 and 1907; it was edited by the geologist and amber specialist Richard Klebs (1850–1911).

This work appeared in a series published by the Prussian Geological Survey in the series *Abhandlungen der Königlich Preussischen Geologischen Landesanstalt*, N.F. 4: 1–181.

Reviewer #3:

Remarks to the Author:

This paper reports beautiful new fungal fossils from the Rhynie chert. The illustrations, particularly the light micrographs, are excellent. This is an important contribution to paleobiology. Nevertheless, I have a number of criticisms regarding interpretation of the fossils themselves and the inferences regarding evolution of plant pathogenicity in Dikarya.

1. The structures projecting from the plant host have been interpreted as conidiophores, which are sporangia that produce asexual spores via mitosis. This is plausible, but there is no direct evidence that the spores were produced via mitosis and not meiosis. I suggest that the authors describe the structures as sporangia, which I do not think will be controversial, and then explain why it is likely that they are mitosporangia. The reasoning, which is not explicitly stated, seems to be that the fossil structures are not clearly asci or basidia (meiosporangia of ascomycetes and basidiomycetes), but why should we assume that meiosporangia of a 400 million year old fungus must resemble those of living taxa?

2. In lines 302-6, the authors write, "We considered but dismissed the possibility that *Potteromyces* is an early representative of the rust lineage of the Basidiomycota. There is some resemblance between our fungus and the spermogonium of a modern rust fungus with receptive hyphae and spermatia, but there are many aspects that do not resemble any extant spermogonia (Aime C. and Duplessis S., personal communication)."

I also thought of rust fungi when I saw these images. The explanation of why this fossil cannot be a rust (or related to rusts) is vague and unsatisfying. Please describe the similarities between *Potteromyces* and the spermogonia of a rust, as well as the precise characters that differentiate them. What exactly are the "many aspects" of *Potteromyces* that rule out a relationship to Pucciniomycotina? It is not enough to cite the unpublished opinions of colleagues. If it is plausible that *Potteromyces* is related to rusts, then its appropriate taxonomic placement is Dikarya incertae sedis, not Ascomycota incertae sedis. This conclusion has strong implications for the section of the ms titled "Dating the phylogenetic tree of Ascomycota".

3. The inference that "pathogenicity first evolved in Ascomycota rather than in Basidiomycota" (lines 372-3) is not well supported. First, it is possible that this fungus is a basidiomycete (see above). Even if it is an ascomycete, the reasoning here is not strong. To estimate the relative dates at which plant pathogenicity first evolved in ascomycetes vs. basidiomycetes would require a combination of molecular clock analyses and ancestral state reconstruction. Ustilaginomycotina and Pucciniomycotina (Basidiomycota) are overwhelmingly composed of plant pathogens. The branching order among Ustilaginomycotina, Pucciniomycotina, and Agaricomycotina is not well resolved, but it is entirely possible that the ancestor of Basidiomycota was a plant pathogen, as was reconstructed by James et al. (2006). In Ascomycota, the Pezizomycotina, which contain the vast majority of plant pathogenic lineages (intermixed with lichens, saprotrophs, mycorrhizae, animal pathogens, and endophytes) are nested within a paraphyletic grade of Saccharomycotina and Taphrinomycotina, of which only the latter includes plant pathogens. The ancestral nutritional mode of Ascomycota is far from certain. The comment that "Extant fungal plant pathogens are mostly found in the phylum Ascomycota and to a very limited extent in Basidiomycota" (lines 358-9) is a red herring. Extant diversity is not relevant to the question of ancestral nutritional modes--it is the diversity that existed in the Devonian that matters here. I see that the study of Lutzoni et al. (2018), which presented a detailed molecular clock analysis of fungi, is cited in this paper. The results of that study could be highlighted in this section ("Lifestyle of *Potteromyces* and the evolution of pathogenicity") with regard to the ages of groups of fungi that contain plant pathogens.

4. I don't see all the details of spore production in the figures that are described in the text, specifically in lines 172 ("Conidia clearly develop at the top of the filaments"), 219-20 ("Conidia appear to develop at the top of the conidiophores") and 252-3 ("...conidia are evidently formed holoblastically and singly at the tip of the conidiophore without any separate septation prior to

secession,..."). In some figures the spores appear along the sides of the sporangia or at the base, and I certainly cannot see the details of spore production. It would be most helpful if the authors could provide line drawings based on the micrographs that show exactly what they are interpreting. This would let readers draw their own conclusions about the structures that are illustrated. The problem here may be my lack of expertise, but it would help if the authors could show me exactly what they are talking about, or in some cases consider being more cautious in their interpretations.

Minor comments:

5. The authors have decided to name the genus after a person, Beatrix Potter. I don't object to this practice in general, but it is controversial. Sometimes fungi are named for an individual who collected the specimen but is not an author of the name. In this case, there appears to be no direct connection between the fossil and the (quite worthy) honoree. I would ask that the authors consider whether a descriptive name might be more informative and useful. This is obviously a matter of preference.

6. The confocal images are beautiful, but I'm not sure I see anything in them that is not visible in the light micrographs. Are they essential? I would rather have line drawings!

In summary, this is an important report of an interesting new fossil fungus from the famous Rhynie chert. As the authors note, there are many other plant-associated fungi in early terrestrial ecosystems, including associations between zoosporic fungi and charophytes (which are almost plants!), fungal endophytes, etc (lines 312-338). The reaction tissue described here does suggest that the host was alive when it was colonized. Nevertheless, the occurrence of other plant-associated fossil fungi, somewhat diminishes the impact of the claim that this is the oldest plant pathogen. I have yet to be convinced that this cannot be a basidiomycete, or that the spores are mitospores. All these criticisms aside, this is a wonderful contribution to the mycota of the Rhynie chert.

Answers TO REVIEWER COMMENTS

We thank the reviewers for their helpful comments. Please find below our answers to these and note that we have modified the text as necessary to address the points raised.

Reviewer #1 (Remarks to the Author):

This manuscript describes a new plant-associated fungus in the Lower Devonian Rhynie Chert. Rhynie is of enormous importance as the oldest terrestrial ecosystem that is preserved with anywhere nears its amazingly complete detail. Any new discovery is worth celebrating and the discoveries keep coming. The authors of this study include some of the most active explorers of Rhynie in recent years and have been pivotal in the continued expansion of our understanding of this ecosystem. This new taxon is a worthy addition to that legacy. The question becomes what to do with it, and there are issues with the context in which the fossil has been placed.

The authors emphasize the significance of the find in two directions: 1., as an early (maybe?) ascomycete that may be useful for time calibration of the fungal phylogeny and, 2., as an oldest fungal pathogen. Neither of those arguments hit exactly right.

Point 1:

The paper ends with a section on time calibration of phylogenies where the conclusion is "Described as Ascomycota incertae sedis, *Potteromyces asteroxylicola* fits into the Ascomycota stem group and could be justified for use as a minimum node age calibration point for the whole Ascomycota (i.e., as the stem node of Ascomycota in a phylogeny) or perhaps even the Dikarya crown group." (Lines 424-426) This feels problematic on two levels. First, so many first appearances stack up at Rhynie, that only the most derived fossils end up mattering for time calibration. In the same section the authors discuss *Paleopyrenomycites* and whether it is stem Pezizomycotina, crown Pezizomycotina, or maybe even within Pezizomycotina, but all of those options are crown Ascomycota.

We share some of the concerns of the reviewer, but we do not see why this prevents any use of *P. asteroxylicola* in time calibration exercises provided they are appropriately employed as an inclusive time calibration analysis should ideally make use of all available fossils and not just those that are most derived.

With regard to *Paleopyrenomycites*, we are not convinced that it can be placed confidently in the ascomycete crown group. The reviewer is correct that *Paleopyrenomycites* has been used in dating fungal phylogeny by various authors (stem Pezizomycotina, crown Pezizomycotina, or maybe even within Pezizomycotina), all crown Ascomycota, but usage does not make something correct. In the text we explain why our fungus cannot be placed with confidence in crown group Ascomycota as certain of its key traits cannot confidently be referred to homologous structures. For example, an operculum in the asci is not clear in the original illustrations. Among extant Ascomycota, operculate asci only occur in apothecial ascomata (i.e., open and disc-like, not covered and with an opening) (Hansen & Pfister, 2006), whereas in this fossil, if it indeed had truly operculate asci, they are associated not with apothecial

ascomata but with perithecial ascomata (i.e., rounded to flask-shaped with a pore-like opening, an ostiole).

Furthermore, perithecial ascomata facilitate forcible discharge of the ascospores (Roper *et al.*, 2008), but the ascospore shape and orientation in *P. devonicus* are not consistent with forcible discharge. Other characters of *P. devonicus* are prone to homoplasy; the ostiole, a distinct apical pore in the ascoma, is known to have evolved independently at least four times in *Ascomycota*. The fossil also has a further suite of features that is not found together in extant *Ascomycota*.

There is therefore considerable uncertainty as to the exact placement of *P. devonicus* and Berbee *et al.* (2020) argue that *P. devonicus* represents an extinct clade of early-diverging *Ascomycota*, a view with which we concur. We do not understand why issues related to the interpretation of *P. devonicus* should preclude any discussion of *Potteromyces asteroxylicola* in the context of time calibration.

So, the new fossil being stem-group *Ascomycota* makes it irrelevant for dating, since *Paleopyrenomycites* is there already in the same rocks and is more derived within *Ascomycota*. There is no calibration the new fossil can offer that *Paleopyrenomycites* can't provide already.

We explain above why *Paleopyrenomycites* cannot be considered as belonging to crown group *Ascomycota* but is better placed in the *Ascomycota* stem-group. *Paleopyrenomycites* and our new fossil *Potteromyces asteroxylicola* are therefore equivalent as calibration points. *Paleopyrenomycites* is not necessarily more derived within *Ascomycota*. Indeed, the presence of two or more distinct ascomycetes in the Rhynie Chert strengthens the use of this fossil site as a calibration point for stem group *Ascomycota*.

Second, there is nothing diagnostic about the new fossil to make it conclusively an ascomycete, and such a fossil simply should not be used for dating.

We have added a new section in the revised version explaining why our fossil cannot be a rust fungus so as to make our reasons for concluding there is an ascomycete affinity clearer.

The manuscript even offers mention of another fossil that can serve as a neutral example that is equivalent. The Late Devonian wood rot described by Stubblefield *et al.* is of great significance for paleoecology but is of no value at all for time calibration because we don't know what that fungus is exactly. Most rots are basidiomycetes, but no diagnostic traits are available, so that rot fossil has been ignored when calibrating the basidiomycete phylogeny, and correctly so. The same logic applies to the new fossil described here and the ascomycete phylogeny. On multiple levels, value for the new fossil as time calibration really isn't there.

While we agree that there is no character (e.g., clamp connection, basidium) in the fossil responsible for the rot in the late Devonian wood that can be attributed to *Basidiomycota*, the situation is different for our new fossil which possesses morphological features of *Ascomycota*. That our fossil can only be placed in the stem group *Ascomycota* does not preclude its use for calibration at that level. For large datasets including all major groups of

fungi, having multiple fossils available for calibrating a particular node can only increase confidence in its reality and narrow confidence limits.

Point 2:

“Our finding also supports an affiliation to Ascomycota rather than to Basidiomycota for the origin of fungal plant pathogenicity.” (Lines 51-52)

“these zoosporic fungi are parasitic, but to date no unequivocal plant pathogen has been described.” (Line 75)

“Zoosporic fungi are common parasites of the Rhynie chert charophyte *Palaeonitella cranii* where they are known to induce cell hypertrophy in the host (Taylor et al. 1992b).” (Lines 314-315)

The distinction between parasitic versus pathenogenic might not be obvious to much of your readership—it was not immediately obvious to me—so you should define it clearly from the outset.

A pathogen is a disease-causing parasite. *Potteromyces asteroxylicola* evidently bursts through the cuticle provoking damage to the plant resulting in reaction tissues directly associated with the fungus and leading to the development of distinctive dome-shaped projections arising through the proliferation of epidermal or subepidermal cells. We added the definition of a pathogen in the introduction to clarify the difference between a parasite and a pathogen for the readership.

As eventually explained, the key distinction appears to be whether or not the plant shows a response, although nothing described here seems as pronounced as the reaction tissue used to verify that putative arthropod herbivory in fossil plants was not post-mortem.

We agree that the reaction we document is not as pronounced as the reactions that have been attributed to herbivory in some Rhynie chert plants. The so-called wound tissues that have been attributed to arthropod feeding are very much larger and more traumatic, often resulting in massive tissue damage. The reaction tissue that we document, which are clearly caused by a fungus, is of a smaller order but nevertheless still a reaction by the plant; the morphological structures depicting this reaction are comparable to those seen in extant fungal pathogens and clearly seen and highlighted in our figures.

But if the distinction being drawn is of a response from the host, then why does the cell distension mentioned for *Palaeonitella* not count? Is it a question of host cell enlargement versus host cell divisions? Is it because you're only after land plants? Wherever the line is being drawn, it feels like a needle is being thread.

We agree that the distension mentioned for *Palaeonitella* counts. But *Palaeonitella* is an aquatic alga and not a plant that lives on land. In this contribution, we concerned with pathogens of embryophytes, plants that are adapted to terrestrial not aquatic environments.

Other things:

“Each enation is associated with a single vascular system” (Line 165)
That should be “associated with a single vein” -- the whole plant only has a single vascular system.

We deleted this sentence as it is not pertinent to our conclusions.

“Etymology: The generic name honours Helen Beatrix Potter (1866–1943)” (Line 192)
I was pleased to learn that Beatrix Potter was a mycologist. Thanks for that.

“Hence, the association with *Lepidocaris rhyniensis* probably happened post mortem when the plant was washed into a shallow pool.” (Lines 342-344)
Or the water levels rose without needing the plant material to be transported?

Reviewer 2 also commented on this point. We made changes to the text and added this possibility.

“Extant fungal plant pathogens are mostly found in the phylum Ascomycota and to a very limited extent in Basidiomycota.” (Lines 358-359)
This doesn't feel fair to Basidios. At 10,000 extant species, rusts and smuts represent a third of basidiomycete diversity. Plus, if you think about the basal topology of the basidiomycete phylogeny, they are one half of the tree.

We thank the reviewer for this comment and have modified the text accordingly with input from a leading rust specialist.

Reviewer #2 (Remarks to the Author):

The authors describe a new fungus from the Early Devonian Rhynie chert that is attributed to the Ascomycota incertae sedis. It is said to be the earliest evidence of a plant pathogenic fungus. Yes, this work will be significant for related fields because it has an impact on our ideas on fungal plant pathogenicity and their origins. The work largely supports the conclusions and claims but some points need to be demonstrated in more detail, i.e. the aseptate nature of the conidiophores and the part on the formation of conidia. Some citations and data cited from the literature are incorrect and need to be corrected. Proper diagnoses should be given. The methods section is fine. Attached an annotated version of the manuscript and a file with comments.

Hans Kerp

[answers provided on the file with comments]

We thank you for your helpful comments that we have addressed below.

This is an interesting manuscript that is definitely worth to be published. However, several points still need attention (see below). Further remarks are made in the annotated pdf file. One of my concerns are the illustrations. They are of good quality, but there are quite a number of duplicates. I really wonder what the many confocal images really add to the excellent light-microscopic pictures. The latter are mostly clearer and suffice. Therefore, I suggest to reduce the number of confocal images to the absolute minimum and retain only those that aspects that are not to be seen or on the LM images.

We followed your suggestion and reduced the number of confocal images by deleting four of them. We consider the remaining confocal images to be essential as they are complementary to the light images giving details at higher resolution. They also form the basis of our 3D reconstructions (see videos).

L. 45. Why “the earliest **unequivocal** fossil evidence of a plant pathogenic fungus”, when the title reads: “The earliest fungal plant pathogen?”. The word “unequivocal” in the text and the question mark in the title seem to be conflicting.

We agree and deleted “unequivocal” from the abstract.

L. 74. Add: Krings, M. *Triskelia scotlandica*, an enigmatic Rhynie chert microfossil revisited. *PalZ* **95**, 1–15. (2021).

We added this reference.

L. 90. replace “early” by “Lower”. Early refers to age or time, whereas Lower refers to a rock series, as is the case here. The adjective Lower must be capitalized because the Devonian is formally subdivided into Lower (Early), Middle and Upper (Late) (see the latest version of the IUGS International Chronostratigraphic Chart). Therefore, it should be “...from the Lower Devonian”; Also correct would be: “The earliest generally accepted fossil lichens are the Early Devonian *Chlorolichenomycites salopensis* and *Cyanolichenomycites devonicus*”. See also correction in l. 367. Please check the use of stratigraphic nomenclature throughout the manuscript.

We made corrections in the text accordingly.

L. 92. “>250 Mya” means more than 250 Myr, i.e. in the Paleozoic!

We deleted >250 Mya from the text.

L. 91. The correct orthography is *Prototaxites taitii* as it was named after Mr. D. Tait, fossil collector to the Geological Survey, who opened the first trenches in the Rhynie field in 1913. For details on the orthography of epithets see Art. 60.8 of the International Code of Nomenclature for algae, fungi, and plants (Shenzhen Code).

This has been corrected in the text.

L. 91. Normally, in taxonomic literature only names of genera and subordinate taxonomic units are printed in italics, and names of higher rank (families, orders, classes, phyla) not. However, this may depend of the style of the journal.

The International Code of Nomenclature for algae, fungi, and plants (of which one of us, David Hawksworth is an editor) does not rule on this matter, but does italicize scientific names at all ranks in the Code. Major mycological journals have been italicising such names for many years, and here we follow the recommendation of the International Commission on the Taxonomy of Fungi: Thines, M., Aoki, T., Crous, P.W., Hyde, K.D., Lücking, R., Malosso, E., May, T.W., Miller, A.N., Redhead, S.A., Yurkov, A.M. and Hawksworth, D.L., 2020. Setting scientific names at all taxonomic ranks in italics facilitates their quick recognition in scientific papers. *IMA fungus*, 11(25), pp.1-5.).

L. 105. Retallack & Landing (2014) interpreted *Prototaxites* as a belonging to the *Glomeromycota*.

Retallack G.J. & Landing, E. Affinities and architecture of Devonian trunks of *Prototaxites loganii*. *Mycologia*, **106**: 1143–1158. (2014)

The reference has been added in the text.

L. 115. Do not abbreviate genus name because this is very confusing, because in the same paragraph you refer to two genus names starting with *P*.

We have amended the generic abbreviations where necessary to avoid confusion.

L. 163ff. It is correct that the enations of *Asteroxylon mackiei* lack vascular tissue. However, the sporophylls are vascularized. It is the question whether the lack of internal vascular tissue in the enations is a primary condition, or whether it is a secondary loss.

We deleted the sentence: « Each enation is associated with a single vascular system, but this is restricted to the cortical zone of the axis on which it develops » as this is not relevant to our findings.

l. 178. See comments to l. 339-344 below.

L. 188. I have some concern with regard to the correctness of the name and the orthography of the epithet. According to Art. 62.2(a) of the Art. 60.8 of the International Code of Nomenclature for algae, fungi, and plants (Shenzen Code), compounds ending in *-myces* are masculine, irrespective after whom the genus is named. The inflection of the epithet should be accordingly. The correct name would thus be: *Potteromyces asteroxylicolum*.

Shenzen Code:

62.2. Compound generic names take the gender of the last word in the nominative case in the compound (but see Art. 14.11). If the termination is altered, however, the gender is altered accordingly.

(a) Compounds ending in *-botrys*, *-codon*, *-myces*, *-odon*, *-panax*, *-pogon*, *-stemon*, and other masculine words, are masculine.

That is indeed the general rule, but the “-icola” suffix is mentioned specifically in the International Code of Nomenclature in Art. 23.5 as one not to be changed. If “-iculum” or “-icolus” is used they are treated as an error and must be changed to “-icola”, This is the pertinent Article:

23.5. The specific epithet, when adjectival in form and not used as a noun, agrees with the gender of the generic name; when the epithet is a noun in apposition or a genitive noun, it retains its own gender and termination irrespective of the gender of the generic name. Epithets not conforming to this rule are to be corrected (see Art. 32.2) to the proper form of the termination (Latin or transcribed Greek) of the original author(s). In particular, the usage of the word element *-cola* as an adjective is a correctable error.

Ex. 11. Correctable error in the usage of *-cola* as an adjective: when Blanchard (in *Rhodora* 8: 170. 1906) proposed *Rubus ‘amnicolus’*, it was a correctable error for *R. amnicola* Blanch

L. 199 ff. I have serious problems with the diagnosis, which is in my opinion not a proper diagnosis. A diagnosis should be a brief characterization of the taxon, highlighting the features in which it differs from closely related and/or similar taxa. Exact measurements do not belong in a diagnosis but in the description. Here measurements are given with a precision of 0.5 µm. What to do with specimens that are morphologically 100% identical but just 0.5 µm narrower or wider? These do not fit the diagnosis! Moreover, when taxa are first described usually little is known about the natural variability, especially in fossil material. Moreover, a combined diagnosis for the genus and species is given. According to the latest edition of the nomenclature code which supersedes all previous editions it is possible to give a combined generic/specific diagnosis. However, it should be noted that such diagnoses are often so tightly defined that it is hardly possible to include additional species. Therefore, I personally prefer separate diagnoses, the generic one being a bit more general than the specific one. Last but not least, according to the current nomenclature rules, it is not strictly necessary to include a diagnosis; a good description will suffice. Nevertheless, I prefer including diagnoses.

We agree and have made changes to the diagnosis and provided separate ones for the genus and the species names.

L. 222-4. Couldn't this be a matter of preservation?

We do not consider that this could be a matter of preservation as the cells are well preserved in each direction (see figure 1C).

L. 253-5. This statement should be further substantiated, because it is not really convincing. Please provide pictures with accompanied explanatory drawings to make this clearer.

We deleted “without any separate septation prior to secession, although the possibility that they are produced by segmentation cannot be entirely discounted as no hyphal walls contiguous with the spore walls were evident”. We explained the conidiogenesis in more detail in lines 216-218 with references to the figures and put arrows on the images to make the statement clearer.

L. 257. Are the conidiophores in *Potteromyces* really aseptate? Figs. 2C and 2D seem to show septa.

The gaps visible in the conidiogenous cells are not septa, if these were septa, the cell wall would have been preserved at the level of the gap.

L. 266. Please note that the term Tertiary is no longer used. It has been replaced by Paleogene and Neogene (see International Chronostratigraphic Chart). Moreover, the ages given here are for Caspary's material incorrect. Kettunen *et al.* (2019. p. 367) wrote:

"These amber-bearing strata are Priabonian (c. 34–38 Ma) in age, but small amounts of amber also occur in Lutetian and Oligocene sediments, leading to a possible age range of c.25–43 million years for all strata bearing Baltic amber (Kosmowska-Ceranowicz *et al.* 1997; Standke 1998; Kasinski & Kramarska 2008; Standke 2008). It is unclear whether the Oligocene amber represents redeposited Eocene material (Standke 2008); thus, a Lutetian to Priabonian age (c. 34–43 Ma) of Baltic amber is currently considered. Baltic amber eroded from these sediments is often found washed ashore along the coast of the Baltic Sea (predominantly in the Baltic States, Poland, Denmark, Germany and in southern Sweden) and in adjacent areas, and a large proportion of historic and new amber collections contains this 'sea amber'. A precise locality of origin therefore cannot be provided for Baltic amber pieces from historic collections that were developed in the Königsberg (Kaliningrad) and Danzig (Gdansk) areas during the nineteenth and early twentieth centuries. This fact does not, however, affect the age estimate given above since the vast majority was initially embedded in these Eocene sediments (Standke 2008)."

Corrected in the text.

L ; 314. Zoosporic fungi are said to be common parasites in the Rhynie chert charophyte. How common?

That was a mistake and thanks for drawing it to our attention. We have corrected this sentence to « a zoosporic fungus is a parasite of the Rhynie chert charophyte ».

L. 328 ff. Sorry, but here I fundamentally disagree! The secondary cell wall thickenings are clearly a response to one particular fungus.

We agree with that statement. There is a response of the plant, however this response is a barrier (a secondary cell wall thickening) formed by the plant and not direct damage caused to the plant by the fungus. Moreover, the affinity of the fungus is not demonstrated. Krings *et al.* suggested that it might be chytrid-like. This is a completely different reaction to the one we describe in our manuscript. We added in our text: « One was described as responsible of

secondary cell wall thickenings however this represents a harmless reaction from the plant against a fungus suggested as a possible chytrid-like ».

I. 339-44. These phrases are a bit confusing. *Asteroxylon mackiei* did not grow in wet environments as the presence of stomata on root-bearing axes demonstrates.

Asteroxylon mackiei is commonly associated by *Horneophyton lignieri*. They grew on sandy, probably well-drained soils. *Asteroxylon* had roots that penetrated the soil. In my opinion, there is no doubt that the association with *Lepidocaris* is post mortem. Probably the plant was washed into a pool. However, it should be noted that *Lepidocaris* fossils are usually exuviae, which are very light and transported very easily, even by wind from ephemeral pools. I suggest to shorten this part to make it clearer.

We agree that *Asteroxylon* grew on sandy substrates but the plant borders shallow pools (Edwards & Selden, 1992). Indeed, one possibility is that the plant was washed into the pool. Another possibility, suggested by Reviewer 1, is that the water levels rose without needing the plant material to be transported. We made changes to the text to accommodate both comments.

L. 350. This is a rather bold statement. Such reaction tissues being evidence for regeneration by Rhynie chert plants have been illustrated before, e.g., Kidston & Lang (1921a), Edwards & Selden (1992), Krings (2021).

We agree that reaction tissues have been reported by several authors, and added the missing references to our manuscript, but we are not aware of reaction tissues directly linked with a fungus with known affinity causing pathogenicity. The reaction we describe is completely different to the reaction tissues previously reported that show the occurrence of cavities in wound tissues with the enlargement and division of neighbouring cells. There is no direct evidence for the cause of these traumatic wound tissues (e.g. Kidston & Lang, 1921, Edwards & Selden, 1992).

As reported by Edwards and Selden (1992), « The wounds show necrosis and hypertrophy of neighbouring cells, the darkening of some cells suggesting a chemical wound response. Various ideas have been put forward to explain the observed features: physical damage by hot water or volcanic ash, by sap-feeding arthropods, or by parasites, fungi, nematodes or mites (Rolfe, 1985)... Fungi might be expected to attack a wound already inflicted, which begs the question of what caused the original wound. »

Other reports show structures linked with cell divisions that might be attributed to various microorganisms (e.g., cyanobacteria, Krings, 2021) or arthropods but, none of them show evidence of a fungus interacting in the vicinity.

L. 372. Callosities have associated with the clamp-bearing hyphae been described by Krings *et al.* (2011, p. 20).

Krings *et al* (2011) suggested that if the callosities formed in response to invading clamp-bearing hyphae, then this host response would favour evidence of a parasitic infection; they

wrote that the fungus could have been biotrophic or saprotrophic, but did not mention the possibility of pathogenicity or necrotrophism. We added this to the text.

L. 464 ff. The references to Caspary are incorrect, respectively incomplete. Both works were published posthumously. The text volume was published in **1906** (not 1907!) and the accompanying atlas in 1907. Robert Caspary was a German botanist (1818– 1887), who was born in Königsberg (now Kaliningrad, Russia). After stays in Bonn, Berlin, Italy, England and France he was appointed professor in botany at the University of Königsberg in 1859. The so-called casparian strips are named after him.

His work on Baltic amber was published some 20 years after his death, in 1906 and 1907; it was edited by the geologist and amber specialist Richard Klebs (1850–1911).

This work appeared in a series published by the Prussian Geological Survey in the series *Abhandlungen der Königlich Preußischen Geologischen Landesanstalt*, N.F. **4**: 1–181.

We thank the reviewer for this helpful information, and we have corrected the dates in the manuscript.

Reviewer #3 (Remarks to the Author):

This paper reports beautiful new fungal fossils from the Rhynie chert. The illustrations, particularly the light micrographs, are excellent. This is an important contribution to paleobiology. Nevertheless, I have a number of criticisms regarding interpretation of the fossils themselves and the inferences regarding evolution of plant pathogenicity in Dikarya.

1. The structures projecting from the plant host have been interpreted as conidiophores, which are sporangia that produce asexual spores via mitosis. This is plausible, but there is no direct evidence that the spores were produced via mitosis and not meiosis. I suggest that the authors describe the structures as sporangia, which I do not think will be controversial, and then explain why it is likely that they are mitosporangia. The reasoning, which is not explicitly stated, seems to be that the fossil structures are not clearly asci or basidia (meiosporangia of ascomycetes and basidiomycetes), but why should we assume that meiosporangia of a 400 million year old fungus must resemble those of living taxa?

We disagree with that interpretation as a sporangium is defined as a structure containing endogenously produced spores where the walls of the spores are not derived from the supporting or containing structure. Sporangia do not occur in *Ascomycota* or *Basidiomycota*. Perhaps the reviewer meant "sporophores" which is a general term that can be applied to any spore-producing structure such as a conidiophore. The structures we observed could be referred to as sporophores, but we interpret them as conidiophores producing mitospores (i.e. conidia) as we do not see how meiospores (i.e. ascospores or basidiospores) could be formed in chains as meiotic not mitotic cell division would be required for each spore.

2. In lines 302-6, the authors write, "We considered but dismissed the possibility that *Potteromyces* is an early representative of the rust lineage of the Basidiomycota. There is some resemblance between our fungus and the spermogonium of a modern rust fungus with receptive hyphae and spermatia, but there are many aspects that do not resemble any extant spermogonia (Aime C. and Duplessis S., personal communication)." I also thought of rust fungi when I saw these images. The explanation of why this fossil cannot be a rust (or related to rusts) is vague and unsatisfying. Please describe the similarities between *Potteromyces* and the spermogonia of a rust, as well as the precise characters that differentiate them. What exactly are the "many aspects" of *Potteromyces* that rule out a relationship to Pucciniomycotina? It is not enough to cite the unpublished opinions of colleagues. If it is plausible that *Potteromyces* is related to rusts, then its appropriate taxonomic placement is Dikarya incertae sedis, not Ascomycota incertae sedis. This conclusion has strong implications for the section of the ms titled "Dating the phylogenetic tree of Ascomycota".

Dr Aime, a leading expert on rust fungi, and co-author of a new edition of an account of the world's genera of rust fungi due to appear shortly, joined us as a co-author to address this issue more fully in our revised manuscript. The following paragraph has been added:

"Within the *Basidiomycota*, only the the spermatia produced within spermogonia (one of five spore stages) of rust fungi (*Puccinia les*) share the character of phialidic spore formation^{46,47} with *Potteromyces*. Where spermatia formation has been studied, spores are formed by basipetal succession from the apex of the spermatophore and then break off to fill the

spermogonial cavity. However, in marked contrast to *Potteromyces*, the spermogonia of extant rust fungi are +/- flask shaped; those of the earliest extant rusts, i.e., *Rogerpetersonia* and *Mikronegeria*, are very deep seated within the host tissue with only the flask necks protruding^{48, 49}. Additionally, the rust fungi themselves are a more recently diverging lineage of *Pucciniomycotina*, estimated to have evolved c. 200 Mya⁴⁸; spermogonia and phialidic spore production is not known from any of the sister lineages to *Pucciniales*, or elsewhere within *Pucciniomycotina*⁵⁰, making it likely that phialidic sporogenesis evolved independently within *Pucciniales* from *Ascomycota*. Finally, *Pucciniales* are biotrophic plant pathogens, whereas *Potteromyces* appears to be necrotrophic, and in extant rust species that form spermogonia, these are accompanied by the formation of a second sorus type, termed an aecium and characterized by production of catenulate chains of spores within a cup-shaped sorus that originates from and is formed in close proximity to the dikaryotized hyphae of the spermogonium. The lack of any analogous structure in *Potteromyces* also makes it extremely unlikely that it represents a species of *Pucciniales*."

3. The inference that "pathogenicity first evolved in Ascomycota rather than in Basidiomycota" (lines 372-3) is not well supported. First, it is possible that this fungus is a basidiomycete (see above). Even if it is an ascomycete, the reasoning here is not strong. To estimate the relative dates at which plant pathogenicity first evolved in ascomycetes vs. basidiomycetes would require a combination of molecular clock analyses and ancestral state reconstruction. Ustilaginomycotina and Pucciniomycotina (Basidiomycota) are overwhelmingly composed of plant pathogens. The branching order among Ustilaginomycotina, Pucciniomycotina, and Agaricomycotina is not well resolved, but it is entirely possible that the ancestor of Basidiomycota was a plant pathogen, as was reconstructed by James et al. (2006). In Ascomycota, the Pezizomycotina, which contain the vast majority of plant pathogenic lineages (intermixed with lichens, saprotrophs, mycorrhizae, animal pathogens, and endophytes) are nested within a paraphyletic grade of Saccharomycotina and Taphrinomycotina, of which only the latter includes plant pathogens. The ancestral nutritional mode of Ascomycota is far from certain. The comment that "Extant fungal plant pathogens are mostly found in the phylum Ascomycota and to a very limited extent in Basidiomycota" (lines 358-9) is a red herring. Extant diversity is not relevant to the question of ancestral nutritional modes--it is the diversity that existed in the Devonian that matters here. I see that the study of Lutzoni et al. (2018), which presented a detailed molecular clock analysis of fungi, is cited in this paper. The results of that study could be highlighted in this section ("Lifestyle of *Potteromyces* and the evolution of pathogenicity") with regard to the ages of groups of fungi that contain plant pathogens.

We deleted in the summary the sentence: "Our finding also supports an affiliation to *Ascomycota* rather than to *Basidiomycota* for the origin of fungal plant pathogenicity". We also deleted in the text, line 528, the sentence: "As the new fungus is evidently a plant pathogen, this supports the view that pathogenicity first evolved in *Ascomycota* rather than in *Basidiomycota*". In the text, we only indicate that our study shows that the earliest fossil evidence for fungal pathogens comes from our ascomycete.

4. I don't see all the details of spore production in the figures that are described in the text, specifically in lines 172 ("Conidia clearly develop at the top of the filaments"), 219-20 ("Conidia

appear to develop at the top of the conidiophores”) and 252-3 (“...conidia are evidently formed holoblastically and singly at the tip of the conidiophore without any separate septation prior to secession,...”). In some figures the spores appear along the sides of the sporangia or at the base, and I certainly cannot see the details of spore production.

We have endeavoured to clarify this by adding arrows on the figures to point out more clearly the formation of the conidia at the top of the filaments (conidiophores). Conidia formed at the tip of the conidiophores is clearly visible in some images (arrows in Fig. 1G, 2B, 3F, 4B). Those that appear along the sides of the conidiophores or at their bases are evidently conidia that have fallen off the conidiophores. There is no direct connection evident between those fallen spores and the conidiophores.

It would be most helpful if the authors could provide line drawings based on the micrographs that show exactly what they are interpreting. This would let readers draw their own conclusions about the structures that are illustrated. The problem here may be my lack of expertise, but it would help if the authors could show me exactly what they are talking about, or in some cases consider being more cautious in their interpretations.

We take the point, but are concerned that line drawings can be misleading as they are necessarily personal subjective interpretations, rather than hard evidence, we have added arrows to the relevant images to draw attention to key features and also included videos in the main manuscript and supplementary material which will allow readers to see the structures in 3D and see exactly what we mean for themselves.

Minor comments:

5. The authors have decided to name the genus after a person, Beatrix Potter. I don't object to this practice in general, but it is controversial. Sometimes fungi are named for an individual who collected the specimen but is not an author of the name. In this case, there appears to be no direct connection between the fossil and the (quite worthy) honoree. I would ask that the authors consider whether a descriptive name might be more informative and useful. This is obviously a matter of preference.

This practice as objected to by some taxonomists but is compliant with the rules and recommendations of the *International Code of Nomenclature for algae, fungi, and plants* (2018), of which one of us (D. Hawksworth) is a co-editor. Moreover, Beatrix Potter was a skilled microscopist who made several novel contributions to mycology but has never been properly acknowledged as much of what she observed was never published. We decided to take this opportunity to do so and consider appropriate to keep the name.

6. The confocal images are beautiful, but I'm not sure I see anything in them that is not visible in the light micrographs. Are they essential? I would rather have line drawings!

We have deleted four of the confocal images. We consider the remaining confocal images to be essential as they form the evidence base for our interpretation and provide 3D reconstructions (see videos) that all can see and assess.

In summary, this is an important report of an interesting new fossil fungus from the famous Rhynie chert. As the authors note, there are many other plant-associated fungi in early terrestrial ecosystems, including associations between zoosporic fungi and charophytes (which are almost plants!), fungal endophytes, etc (lines 312-338). The reaction tissue described here does suggest that the host was alive when it was colonized. Nevertheless, the occurrence of other plant-associated fossil fungi, somewhat diminishes the impact of the claim that this is the oldest plant pathogen. I have yet to be convinced that this cannot be a basidiomycete, or that the spores are mitospores. All these criticisms aside, this is a wonderful contribution to the mycota of the Rhynie chert.

We agree absolutely that there are other plant-associated fossil fungi described, however most of them are probably saprotrophic. Indeed Kidston & Lang (1921) described 15 fungi as saprophytic or possibly mycorrhizal. Subsequently, a few parasitic associations have been described – Reviewer 2 asked the question about our use of “common” concerning the fungus colonizing *Palaeonitella*. That was a mistake, as only one fungus was observed – there are many fungi associated with plants in the Rhynie chert, but the nature of the association (i.e., saprotroph, parasite) is typically inconclusive. We mention the better documented associations in our text and explain how they are different to the pathogenic association we describe.

Reviewers' Comments:

Reviewer #1:

Remarks to the Author:

I think this revision was helpful. Regarding the value for time calibration, I wasn't convinced by the counter-arguments from the authors as presented in the response letter, but I think they did a better job in the actual manuscript. The one thing I might suggest is to look a little more broadly in the citations of that section. For example, I think it is the arthropod world where there has most been a push for establishing standards for fossil calibration. Once you throw out all of the fossil spiders that kind of look like a particular family and instead rely only on specimens where you can verify key defining synapomorphies for the group, then that can have a big impact on the results for dating. You might want to add in some reference to the work of Joanna Wolfe and company (e.g., Wolfe et al. 2016 in Earth-Science Reviews) as example of where this trend is going even if it might take a while to reach the fungi. So, I still question the likelihood that this new specimen is likely to prove helpful in that context in the future given all the other material of the same age from Rhynie, but I think that issue is at least treated with an appropriate amount of ambiguity now and don't see a need to dig in on the topic.

And I like the Beatrix Potter stuff even more now that authors had to fend off the other reviewer with more detail.

Reviewer #2:

Remarks to the Author:

Please see my review of the first version of this manuscript. All queries have been answered and remarks have been dealt with. I recommend publication of the revised version of the manuscript.

Reviewer #3:

None

Answer to Reviewer 1's comments

R1 comments

I think this revision was helpful. Regarding the value for time calibration, I wasn't convinced by the counter-arguments from the authors as presented in the response letter, but I think they did a better job in the actual manuscript. The one thing I might suggest is to look a little more broadly in the citations of that section. For example, I think it is the arthropod world where there has most been a push for establishing standards for fossil calibration. Once you throw out all of the fossil spiders that kind of look like a particular family and instead rely only on specimens where you can verify key defining synapomorphies for the group, then that can have a big impact on the results for dating. You might want to add in some reference to the work of Joanna Wolfe and company (e.g., Wolfe et al. 2016 in Earth-Science Reviews) as example of where this trend is going even if it might take a while to reach the fungi. So, I still question the likelihood that this new specimen is likely to prove helpful in that context in the future given all the other material of the same age from Rhynie, but I think that issue is at least treated with an appropriate amount of ambiguity now and don't see a need to dig in on the topic.

Response:

We made a change (highlighted in yellow) to the following sentence to address the reviewer's comment:

Although various analytical methods try to deal with and reduce the uncertainty associated with the phylogenetic placement of fossil calibration points, incorrect fossil ages (e.g., ⁶⁹⁻⁷¹), and the **use of fossils without key defining synapomorphies for the group**⁷³, the reality is that an erroneously placed fossil, due to a wrong interpretation, may result in a cascade of misleading conclusions about the evolution of *Fungi* and their interactions with other organisms.

And we added reference 73: Joanna M. Wolfe, Allison C. Daley, David A. Legg, Gregory D. Edgecombe, Fossil calibrations for the arthropod Tree of Life, Earth-Science Reviews, 160, 2016, 43-110.